# How Tibetan Nuns Become Khenmos: The History and Evolution of the Khenmo Degree for Tibetan Nuns

## Padma'tsho (Baimacuo)

Department of Philosophy, Southwest Minzu University, Chengdu 610041, China; padmatsho@icloud.com

**Abstract:** Tibetan Buddhist nuns are making history in numerous ways. They now meet in classrooms instead of tents, earn the title "Khenmo" after many years of dedicated study, and take exams that are standardized, frequent, and both written and oral. Additionally, the new educational system encourages Tibetan *Jomos* to take on more responsibility, increase their scholarship and practice, and obtain superior monastery/nunnery status. This article chronicles over two and a half decades of extensive fieldwork, covering the advances in monastic education and the rising standing of women in Larung Gar and contemporary China. These advances are in stark contrast to the limited opportunities for women in the past.

**Keywords:** Tibetan nuns; Khenmo degree; history; evolution; Larung Gar

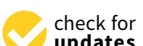



## 1. Introduction

There have been significant developments in contemporary Tibetan Buddhist culture, including the advocacy of vegetarianism and non-violence and the enhanced and increasingly rigorous nuns' education. As these changes have taken place, there are several areas of growth to note and study. For example, two volumes of writings by remarkable women about their lives have been published by monastic nuns at Larung Gar[1], representing a new level of outreach and promotion of Buddhist women's history. There has been a significant increase in both men and women Chinese-Tibetan Buddhists[2]. Many modern Tibetan Buddhist Lamas[3] use the internet and travel to establish centers in China and globally for study and practice, and monastic monks have built numerous schools for villagers. Of all these developments, arguably, the most important one is the new status of Khenmo[4] for Tibetan Buddhist nuns. This present-day role for nuns in Tibetan Buddhist history follows a rigorous, lengthy, and desirable educational curriculum and is available to Tibetan women at Larung Gar. Khenmo is a new word in Tibetan culture and means a female Buddhist teacher who has met the scholastic qualifications, in contrast to the title of Khenpo[5], which is a male Buddhist teacher.

With the writing and publication of the research articles[6] "The Status and Role of Contemporary Tibetan Buddhist Nuns: The Case of Yachen" (Baimacuo 2011) in Chinese and "Courage as Eminence: Tibetan Nuns at Yachen Monastery in Kham" (Baimacuo 2014, pp. 185–94). in English, other related research questions arose. How did the Khenmo degree develop? What courses do they teach? When did the Khenmos begin to teach the five major subjects of exoteric Buddhist study (*gzhung po ti lnga*)[7]? What curriculum do they use? How many monastic nuns have had the opportunity to study Dharma? What do they study? Finally, what are the differences between what nuns and monks learn? To answer this lengthy but vital list of questions, "An Investigation of a New Educational System for Contemporary Tibetan Nuns" (Baimacuo 2015, vol. 3, pp. 176–83) was published. It covered the process of earning the Khenmo degree at Larung Gar and assessed the effects of this new role of Khenmo for Tibetan nuns in Tibetan society and history, and the fieldwork has continued up to the present.

In the past, there were no nunneries or public settings in Tibet that offered comprehensive teaching curricula on the five major subjects of exoteric Buddhist study for *Jomos*[8]. Since the Cultural Revolution, Tibetan Buddhism has slowly recovered in Tibetan cultural regions of China, and many of the 6000 plus destroyed monasteries have been rebuilt. During the first ten years after the Cultural Revolution, monks had many opportunities to study, but similar institutions for nuns were non-existent.

During 1995–1997 and in 2000, extensive fieldwork began at Zungchu[9] nunnery in Aba Prefecture, Lho Dzong nunnery[10] in Jianzha Tshosnon, and Ani Tshankhung nunnery[11] in Lhasa, all considered to be heartlands of Tibetan culture in China. It was found that most monastic nuns just sat around much of the day, chatting. Some studied meditation, but they did not have access to an educational system to explore the higher Buddhist curriculum as the monks did. Then, during the development of the monasteries of Larung Gar and Yachen Gar[12], many nuns came to the two monasteries and, with the innovation and support of Khenpo Jigme Phuntsok (*mkhanpo' jigs med phun tshogs*)[13], helped to change the educational system of Tibetan Buddhism for nuns. This change involved advanced education and training being given to the nuns and those recognized attaining the title of Khenmo in Yachen. This change in Yachen occurred after Khenpo Jigme Phuntsok gave suggestions to Lama Achos. Khenpo Jigme Phuntsok started the new Tibetan Buddhist philosophy for women in the Tibetan area, and slowly created a comparable education system for Tibetan nuns as that for the monks.

Larung Gar in Sertar and Yachen Gar in Palyul are two representative and important nunneries in Eastern Tibet. Although the two monasteries' histories are relatively brief, thousands of monks and nuns live and study Buddhism on site. These institutions have transformed from traditional monasteries into two vast and dynamic Buddhist teaching institutions, widely known among Tibetan and Chinese Buddhists. Yachen and Larung Gar look similar from the outside, but they are distinctive on the inside. Regarding Tibetan Buddhist institutions, there are significant differences between a philosophy teaching center (*bshad grwa*) and a practice center (*sgrub grwa*). The primary function of Larung Gar is that of a Dharma and Buddhist philosophy teaching center (*bshad grwa*), whereas Yachen is the principal place to practice the esoteric forms of Great Perfection (*rdzogs pa chen po*) (Baimacuo 2014, pp. 185–94), and has the function of being a retreat center "*sgrub grwa*" (Baimacuo 2011, vol 3, pp. 165–72). Larung Monastery offers studies in Buddhist philosophy and the five major subjects of exoteric Buddhist study. In contrast, at Yachen, they teach the tantras and Great Perfection, including the development stages and dissolution phases of Great Perfection. In essence, regarding the two monasteries and what they teach, we have the two different approaches/paths to Enlightenment, that of the scholar and that of the yogi practitioner. These two monasteries were built in the 1980s and started teaching Dharma and meditation practice to a young generation of Tibetan boys and girls. Though the boys had considerably more opportunities for these teachings, it was still historical when the Lama offered these teachings to girls. Traditionally, monks were the only ones to receive instructions on the philosophy of Dharma and the tantric practices in public settings. Teaching some of this curriculum to Tibetan nuns launched a change in Tibetan Buddhist education and represented a remarkable moment in Tibetan history. This change and the title of Khenmo started in Larung Gar; therefore, this article will focus on how Tibetan nuns became Tibetan Buddhist teachers and Khenmos in Larung Gar, and is based on many years of interviews and fieldwork[14].

## 2. Breaking the Traditional Structure and Setting Up the Khenmo Degree

In Tibetan history, many famous Lamas have written numerous books and commentaries regarding the exoteric and esoteric forms of Buddhism. There are so many high Lamas[15] who can give teachings and empowerments that the list is almost countless. Resultantly, there are many highly educated *dge bshes*[16], Khenpos, and reincarnated Lamas (*tulkus*) for monks to look up to, but few exemplars for nuns. The primary two analogous female models are Yeshe Stogyal and Machig Labdron.

However, nuns' education has now changed with the developments at Larung Gar and Yachen. Changes started in the mid to late 1980s, went into the 1990s, and have continued until now, and nuns now receive full Dharma teachings, including tantric practices. The nuns' status has dramatically improved because they now study the five major subjects of exoteric Buddhist study and Buddhist philosophy. Exceptional nuns earn the title of Khenmo, which means they are recognized as a female Buddhist teacher/scholar, just like the Khenpo title for males, and they can give official and exalted teachings to nuns. Khenpo Jigme Phuntsok instituted this change in 1990. He passed away in 2004, but his new approach has been carried on and supported by many Khenpos, especially Khenpo Tshul khrims Blo Grus (Khenpo Tsultrim Lodro)[17].

For more than ten years, the author has been researching the nuns at Larung Gar and discussing the education system and the status of Khenmo. When we look at the history of allowing, developing, and awarding the Khenmo degree, it has gone through several stages, the first of which the author refers to as the Initial Transformation stage. The Initial Transformation stage of the Khenmo system refers to when Khenpo Jigme Phuntsok gave the title of Khenmo to *Jomos* in 1990.

Khenpo Jigme Phuntsok established Larung Gar in 1980 and allowed *Jomos* to study and practice Tibetan Buddhism in more depth. In the early years, *Jomos* such as Jetsuma Mumtso[18] arrived at Larung Gar to train with the Khenpos. After a few years, more *Jomos* came and stayed on to study Buddhism, and this development led to Tibetan Buddhist nuns having further opportunities to learn. The author has interviewed Khenmo Chos dbang (Cho Wang)[19] since 2010, and she recounted to the author her arrival at Larung Gar in 1987 to study Buddhism and the Dharma. She remembers when Khenpo Jigme Phuntsok said, "You are the 95th nun here!" Khenmo explained that in 1990 there were 700 nuns, an increase of 605 nuns in only three years.

When the author asked Khenmo Chos dbang when and why Khenpo Jigme Phuntsok gave the title of Khenmo to nuns, she recalled that the Lama[20] said, "There was no Khenmo before, but to improve women's status, it would be excellent if in the future nuns could teach, so we need to set up Khenmo." For the first time, the Lama gave the title of Khenmo to six *Jomos*[21] in 1990. Another Khenmo related when the Lama[22] said, "Because Tibetan women are humble and weak (*nyams chung*), and they don't have their own monasteries, thinking of the future situation of women I will give the title of Khenmo to nuns."[23]

Based on the clear statements of several Khenmos as noted above, as well as many years of research at Larung Gar regarding Khenpo Jigme Phuntsok's motivation for establishing the program, this raises areas of question and concern I have regarding some of the contents of the article by Liang and Taylor (2020) "Tilling the Fields of Merit: The Institutionalization of Feminine Enlightenment in Tibet's First Khenmo Program". For example, information regarding exam semesters, daily schedule, and debate exam requirements/style that they cited was for Han Chinese nuns and not Tibetan, and a differentiation between these was not made. See Footnotes 29, 38, 47, 48, 49, and 55 for more details. Also, there was a considerable difference in the financial situations of Chinese and Tibetan students, which is relevant to a key point. Chinese students have been known to frequently bring considerable financial resources to Tibetan monasteries and institutions, including Larung Gar, since the mid 1990s. Chinese monies were donated to Larung Gar and used to build the hall, classrooms, library, and hotels, which were originally used by the Chinese and are now used by all. My main concern is the primary tenet according to Liang and Taylor (2020) regarding the elevation of the nuns' educational opportunities and status being tied to increasing the nuns' value and source as a field of merit and consequently leading to an increase in the donations that would be offered to them, thus significantly increasing the money coming to the institutions. During an online forum, this author had a personal conversation with the male author regarding this very point, and he indicated that this was in fact true. When asked about the source of his information, he indicated that it was from personal, private conversations, but he offered no examples or sources. I find it most unusual that in my 20 years of research and personal relationships with numerous

Khenmos, and even the most senior Khenpos, none of this was ever mentioned, even in passing. In fact, on several occasions, numerous Khenmos and senior Tibetan nuns related to me conversations they had with Khenpo Jigme Phuntsok regarding the importance of these changes in education for nuns to improve their knowledge and wisdom, and to help their fellow Tibetan nuns do the same. Consequently, I wonder if their main tenet is more of a Chinese idea and motivation, as it is in direct conflict with that of the Tibetan sources.

To recognize and encourage the nuns to study, Khenpo Jigme Phuntsok gave six nuns the title of Khenmo on 1 November 1990, according to the Tibetan calendar[24]. Resultantly, female Buddhist teachers, just like male Khenpos, were allowed to give teachings to nuns. Jetsuma Mumtso was one of the six, and she is the main Lama[25] for the nuns and monks of Larung Gar, where she provides the empowerment of Vajrasattva since Khenpo Jigme Phuntsok passed away. So, a new role for women in Tibetan Buddhism was created by Khenpo Jigme Phuntsok after he visited Nepal, India, and Bhutan from January to March 1990. Khenmo Chos dbang got her Khenmo degree in 1991 after passing the oral exam. The exam included explaining Madhyamika and reciting a few sections of the primary texts. Now the Khenmos exam has become more difficult and challenging, so Khenmo Chos dbang told the author, "Compared to now, we didn't have an exam at the beginning." Before Khenmo Chos dbang went to Setar, she started to study Tibetan with her uncle when she was 13 years old, while she helped the family to tend cattle in a pasture in Brag'go (Luhuo) county, Kham. She became a nun when she was 18 years old and studied with Khenpo Jigme Phuntsok at Larung Gar. She is an exemplary Khenmo who became Director of the Education Division for Tibetan nuns and one of the heads of Si Guan Hui[26] in Larung Gar. However, there was no structure for Tibetan nuns to study the Dharma and Buddhist philosophy until Khenpo Jigme Phuntsok created it. When Chos dbang, a certain exceptional nun, went to Larung Gar and met Khenpo Jigme Phuntsok, he encouraged her to study, advance as a female scholar, and become a Khenmo, to give women practitioners more exceptional opportunities to learn and progress. Her becoming a Khenmo was a monumental change for Tibetan women. Khenmo Chos Dbang remembers that Khenpo Jigme Phuntsok told nuns, "Khenmo and *Ged Longma* (Bhikkhuni) are the same. The *Ged longma* lineage was lost, and we just have *Ged Tshulma* (Samanera) now. Nevertheless, to progress the education level of nuns and be of benefit to them, we gave you the title of Khenmo." Awarding the Khenmo degree has altered the Tibetan Buddhist education system and unsettled traditional Tibetan society. The following are examples of nuns' opportunities to become a Khenmo, or function as one. The author began interviewing Tibetan nuns in 2010, and they told the author that if they studied hard and did well, they would have a chance to be a Khenmo. They were very excited about this. Sadly, one of the nuns told the author that she studied very hard and wanted to be a Khenmo and empowered to give Buddhism teachings. Nevertheless, because she got sick and had to go to Chengdu to see a doctor, she lost her chance to study and become a Khenmo. She was so disappointed and still goes to the doctor in Chengdu annually for her illness. When asked if she preferred to live in a monastery or a big city, like Chengdu, she was adamant that she preferred living in the monastery. It is not just men who want to become Khenpos, but women also want to accomplish Khenmo level and improve their standing. Many nuns in Larung Gar told the author they wanted to study Dharma and benefit other beings like Khenmos do. To have the same aspirations and rights as Khenpos do and to practice Buddhism is their heart's desire. There are many works in *Gangkar Lhamo* ((Baimacuo) 2021) regarding nuns wanting to study culture and Buddhist philosophy. As a Tibetan nun wrote, "Due to the kindness of the Buddha and *Lamas*, we have the opportunity to be of service to the Buddhist teachings and our nationality, and it is the responsibility of all nuns from the three Tibetan regions to do so. Today there are educational opportunities that were impossible in the past. Because of this, everyone should work hard to study. As a young *Jomo* myself, I have little learning. Still I hope that in the future I can write more about my own thoughts and sorrows. For this young nun, serving the cause of equal rights for women while practicing and studying Buddhism, is my sole mission and calling."

(Bde chen dbyangs skyid 2013) This educational opportunity for nuns represents a positive trend towards gender equality in Tibetan Buddhism ((Baiamcuo) and Jacoby (2020)), since its inception in the 1990s.

### 3. The Development of the Khenmo System

The Khenmo system has matured and has become more rigorous over the past 28 years. The author sees three different developmental periods, or stages, of the Khenmo system with this ongoing fieldwork. The three developmental periods are the Initial Transformation, Reconfiguration, and Enhancement and Refinement.

### 3.1. The Initial Transformation of the Khenmo System

The Initial Transformation was revolutionary for Tibetan nuns and women because it redefined Tibetan Buddhist education history and Tibetan women's history. As one Khenmo told the author, Khenpo Jigme Phuntsok sometimes said that the most significant work he did in the world was setting up the nunnery and giving the title of Khenmo to nuns, an extraordinary statement from such an accomplished and high *Lama*. He started this history of Khenmo and provided the opportunity to all Tibetan women and nuns. The Initial Transformation period took place from November 1990 and ended in January 2004 when Khenpo Jigme Phuntsok passed away.

The primary differences during the other two periods were developing the curriculum and standardization of exams for Khenmos. In the 1990s and early 2000s, when Khenpo Jigme Phuntsok gave the title of Khenmo to nuns, some exams were just the simple recitation of a few paragraphs of scriptures or commentaries, and included answering or explaining a few questions from Prajnaparamita (*phar phyin*), Madhyamika (*dbu ma*), Abhidharma (*mngon par chos, mdzod*), and Tantra Section (*rgyud sde*). Chos dbang was appointed as a Khenmo in 1991 by Khenpo Jigme Phuntsok after she passed the exam with her explanation of Madhyamika and recited the textbook of the *Madhyamaka-lamkara* (*dbu ma rgyan gyi rnam bshed*), the *Natural State of the Mind* (*sems nid ngal gsol)*, and the *Jewel Treasure of the Dharmadhatu* (*yod tan mdzod*). The exam took a few days, and she had to explain and recite whatever Khenpo asked in front of five Khenpos. The examiners were Khenpo Jigme Phuntsok, Khenpo Sodargye, Khenpo Phurgrug, Khenpo Rakhog, and Khenpo Dedpa. Khenmo Bzang po (Zang bo) told the author that she recited and verbally explained some paragraphs of the *Bodhicharya Avatara* (*Spyod 'jud*), *Relaxing in the Natural State of the Mind* (*sims nid ngal gsol*), and the *Essence of Luminosity* (*'od gsal snying po*) before she achieved the Khenmo title in 1997. Khenmo Rin 'dzin (Ren Zen)[27] told the author she also became a Khenmo in 1997. The exam was different from the current one, but the essential criteria (*tshad gzhi*) were there. She told the author that in the early days as a Khenmo, she needed to study as much as possible and genuinely understand the Dharma. Then she offered teachings, helping a class or students who wanted to learn the Dharma and Tibetan reading and writing. Additionally, she observed the three disciplines (Gser thang bla rung nang rig nang bstan slob gling 2017, pp. 5–8)[28] of unity, pure perception, and diligence in study and practice. As the *Lamas* have continuously refined and elevated the scholarship of the Khenmos, the second group was more advanced than the first, and the third was more advanced than the second, as is often the case in succeeding developments. During the Initial Transformation period, Khenpo Jigme Phuntsok awarded and conferred the title of Khenmo on five different occasions (See the Table 1).

**Table 1.** Khenmos during Khenpo Jigme Phuntsok's lifetime.

| Year | Time | Number of Khenmos |
|------|------|-------------------|
| 1990 | First | 6 |
| 1991 | Second | 10 |
| 1994 | Third | 19 |
| 1996 | Fourth | 4 |
| 1997 | Fifth | 5 |

After the first six Khenmos earned the title in 1990, ten nuns followed in 1991, and nineteen assistants[29] became Khenmos in 1994. Four assistants became Khenmos in 1996, and five assistants became Khenmos in 1997 after recitation and answering questions in front of a few Khenpos. There were five occasions when forty-four Khenmos earned their title from Khenpo Jigme Phuntsok. Of these, two Khenmos have passed away, and two Khenmos returned home as laypeople. The important concern here is there were very few Khenmo degree exams during this period. The first Han Chinese Khenmo[30] was named in 1991 alongside Tibetan Khenmos.

### 3.2. The Reconfiguration of the Khenmo System

Reconfiguration followed after Khenpo Jigme Phuntsok's passing, with further enhancement and refinement of the criteria for nuns to become Khenmos. Reconfiguration began in 2005, continued to 2012, is still ongoing. It started with formal classes in Buddhist philosophy, taught by Khenmos. Beginning with Khenpo Jigme Phuntsok and furthered with the encouragement of Khenpo Tshul khrims Blo Grus, there were a few Khenmos ordered by Larung Gar to start teaching the five major subjects of exoteric Buddhist study. In 2005, four Khenmos were teaching Madhyamika (*dbu ma*), one Khenmo was teaching Vinaya (*'dul ba*), one Khenmo Epistemology (*tshad ma*), one Khenmo Gateway to Knowledge (*mkhas pa'i tshul la 'jug pa'i sgo zhes pya ba'i bstan bcos*), and two Khenmos the Way of the Bodhisattva (*byang chub sems dpa'i spyod pa la' jugs pa bzhugs so*). It was a historical moment for the Tibetan Buddhist education system (Baimacuo 2015, pp. 158–63) when Khenmos began teaching formal classes of Buddhist philosophy to nuns. Before 2005, the Khenmos' role, as required at Larung Gar, was that of merely acting as teaching assistants for the Khenpos.

According to findings in the author's fieldwork, before 2005, a few Khenmos gave private instruction of readings and writings of *Blo spyong*[31], and some Buddhist philosophy, to nuns requesting such. When the author asked Khenmo Rin 'dzin, "Were there Khenmos teaching Buddhist philosophy before 2005?", she said, "I came to Larung Gar in 1996 when the *Lama* was still living, and there were a few Khenmos who gave some teachings to students who asked. My teacher, Yang gcis (Yang ji), who is more than 50 now, taught *Blo spyong* (mind training), *'dul ba* (Vinaya), *Dbu ma* (Madhyamika), and *Mngon par mdzod* (Abhidharma), but she did not give the teachings formally or to an entire class. She had about ten students, and after some time, there were just three of us left." Nuns could study Tibetan writing and reading and Buddhist philosophy from female teachers before 2005. However, Khenpo Jigme Phuntsok opened the education gateway for nuns and gave them a venerated position, and then there were further developments by Khenpo Tshul khrims Blo Grus and other Khenpos. The Han Chinese nuns' education system is different from that of Tibetan nuns. The Han Chinese nuns began the five major subjects of exoteric Buddhist study in around 2003, taught by Khenpo Sodargye in Chinese, and this has continued until now. Han Chinese nuns are divided into different classes according to the five major subjects of exoteric Buddhist study and Pure Land practice classes, versus Tibetan nuns, who are divided into different classes according to which Khenmos the nuns want to follow. The Han Chinese instructor Khenmos conduct class tutoring after Khenpo Sodargye's teaching. They have two big written exams each year, and a small written exam every month. Each class selects students to conduct oral tests and recitation tests every day. The exam on the five major subjects of exoteric Buddhist study for Han Chinese started in around 2005. Whether they can become a Khenmo instructor is based on their test and exam scores, character and behavior, and the master's[32] opinion. In an interview with Khenmo Zhao wu in 2013, she said: "The length of study for nuns is 6 years in Larung Gar. Afterwards, some of them are invited by the school to stay as class instructors, and some go back to their home monasteries to teach classes, according to the requirements of the monastery."

The number of formal classes taught by Tibetan Khenmos grew during the Reconfiguration period. Tables 2 and 3 show the number of Khenmos and the curriculum in different years.

**Table 2.** The number of Khenmos and the curriculum from 2006 to 2009.

| Year | Class | Number of Khenmos |
|------|-------|-------------------|
| 2006–2007 | Culture | 3 (at least) |
|  | Tibetan Medicine | 1 |
|  | Vinaya | 2 |
|  | Epistemology | 2 |
|  | Abhidharma | 2 |
|  | Madhyamika | 2 |
|  | Prajnaparamita | 2 |
| 2008–2009 | Culture | 2 (at least) |
|  | Tibetan Medicine | 1 |
|  | Vinaya | 3 |
|  | Epistemology | 2 |
|  | Abhidharma | 2 |
|  | Madhyamika | 2 |
|  | Prajnaparamita | 2 |

**Table 3.** The number of Khenmos and the curriculum from 2010 to 2011.

| Year | Class | Number of Khenmos |
|------|-------|-------------------|
| 2010–2011 | Culture | 5 |
|  | Tibetan Medicine | 1 |
|  | The Way of the Bodhisattva | 4 |
|  | Vinaya | 3 |
|  | Epistemology | 3 |
|  | Abhidharma | 3 |
|  | Madhyamika | 3 |
|  | Prajnaparamita | 3 |

Some Khenmos' assistants became Khenmos in 2009 and began teaching classes in 2010. So, the numbers steadily increased. For example, the study of culture added two more courses in 2011. The courses on culture are *Rigs gnas* in Tibetan, and they mainly teach about Tibetan grammar, literature, and poetics.

There were forty-four Khenmos in 2004, but not all taught formal courses at Larung Gar in 2005. Most Khenmos were assistants for other Khenmos' and Khenpos' classes. In November 2009, twenty-eight nuns passed the *Abhidharma*, *Madhyamika*, and Prajnaparamita exams, and twenty-five of them taught formal courses. Still, they did not have the official certification of Khenmo until 2013. According to an Education Management interview[33], 55 Khenmos were teaching official courses from 2010 to 2016. About one hundred assistants were helping with the Khenmos' teaching (Baimacuo 2015, pp. 158–63).

After Khenpo Jigme Phuntsok passed away in 2004, twenty-eight nuns passed the Khenmo exam in November 2009. The exams were different and more demanding than before because nuns in Larung Gar had been studying the five major subjects of exoteric Buddhist study, and the philosophy of Buddhism, and each course of study had an exam each semester. These twenty-eight nuns received the highest grades when they finished each of the *Gzhung po ti lnga*. The exams included two parts, oral explanation ('*chad rgyugs*) and recitation (*blo rgyugs*). When the author interviewed Khenmo Yon tan[34] in July 2013, the author asked, "What kind of exams did you take for the Khenmo degree?", and she told the author that she passed each exam of the five major subjects of exoteric Buddhist study after each course. She took the exam for Abhidharma a few years ago, the one for Prajnaparamita about five years ago, and the Madhyamaka exam six years ago. Then,

Khenpos and Khenmos selected and appointed them as Khenmos three years ago. She taught Abhidharma for two years and Vinaya for two years. So, Khenmo Yon tan started to teach formal philosophy classes in 2010.

### 3.3. The Enhancement and Refinement of the Khenmo System

The Enhancement and Refinement of the Khenmo system meant standardizing the exams and the curriculum of Khenmos after 2013. The academic environment changed significantly. The nuns' classrooms were moved from tents[35] to the new education building, with about 60 classrooms. The educational requirements for nuns became clear and forthright[36]. Table 4 states the nuns' opportunities to study the Dharma with female teachers emerged and steadily increased owing to the fact of there being twenty-eight titled Khenmos.

**Table 4.** The numbers of Khenmos who taught the main formal courses in 2013.

| Course | Number of Khenmos | Course | Number of Khenmos |
|---|---|---|---|
| Culture | 4 | Prajnaparamita | 6 |
| Vinaya | 8 | Madhyamika | 10 |
| Abhidharma | 4 | Epistemology | 8 |
| Mind Training | 3 | Tantra Section (Rgyud sde) | 5 |

Compare this with the education situation before the Larung Gar institution expanded the number of classes in 2013 because of more titled Khenmos. The numbers of formal courses in Table 5 are the numbers of Khenmos who taught those courses in 2013. However, a Khenmo from Education Management reported that there were forty-eight Khenmos teaching courses. During this period, they started the Tantra Section (*rgyud sde*) class for nuns because an old Khenmo and Khenpo Chos Phan asked. The Tantra Section class looks like the traditional contemplative track of Buddhist training for nuns, but in addition to meditation, nuns also need to study and take exams. The Tantra Section class is generally five years; three years are contemplation, and two years studying the four mind-changings (*Blo ldog rnam bzhi*). It is mainly about the training and learning of renunciation, Bodhicitta, the meaning of esoteric Buddhism, and guru yoga. The exams for the students in the class are taken once a year, and they ask students to explain the seven Dharmas of Vairochana (*rnam snang chos bdun*) in terms of body, speech, and mind. If nuns want to continue after five years of study, it will take at least another two years to complete Dzogchen. There were about 80–100 nuns in the class when I was in Larung Gar in 2013 and 2017. This means that most nuns in Larung Gar were in the classes of the scholastic system to study culture and the five major subjects of exoteric Buddhist study. Many Tibetan nuns told me that they came here because they did not have opportunities to study in their hometown nunneries. A Tibetan nun wrote a work of prose, "Our Rare Time", and said, "In particular, for women who face numerous internal and external challenges, we have the chance to encounter rare opportunities. We need to seize the moment and strive for the pursuit of knowledge. This is an appropriate undertaking that need not feel burdensome. If we have courage, confidence, knowledge, and altruism, nothing can stand in our way. With knowledge, we can discern which activities to undertake and which to relinquish."[37] Studying Dharma and becoming a Khenmo is the purpose for most nuns in Larung Gar.

**Table 5.** The number of Khenmos who taught the main formal courses in 2017[38].

| Course | Number of Khenmos | Course | Number of Khenmos |
|---|---|---|---|
| Culture | 6 | Prajnaparamita | 4 |
| Vinaya | 6 | Madhyamika | 3 |
| Abhidharma | 8 | Epistemology | 5 |
| Mind Training | 5 | Tantra Section | 3 |

Now nuns and Tibetan women, mostly known for reciting texts, have access to Buddhist philosophical teachings and the highest tantric practices. Since 2013, the education system has steadily improved by clarifying the requirements, increasing the number of exams and standardizing them for the Khenmo degree, and establishing the curriculum for Khenmos to teach. The classroom environment has improved significantly, as well as the opportunity to study in greater depth. The Khenmos' teaching system has improved substantially with the addition of more discussion and exams. A nun's rank determines her study subjects, as a beginner, intermediate, or advanced student. Nuns choose for themselves the section and classes to attend when they finish the Culture section. They have been using the same textbooks as the monks. From interviewing over 100 nuns over the years, the author learned that some studied for 5–10 years with one Khenmo, and when asked about the daily schedule of Tibetan nuns, learned that they stay quite busy all day. Tibetan nuns[39] get up before 05:30 to chant and read books; at 08:00, many study *The Words of My Perfect Teacher*; from 09:30 to 11:30, the Khenmo teaches; at noon, they eat lunch; and from 13:00 to 13:30 they fulfill prayer requests. From 13:30 to 14:30, the Khenmo's assistant takes questions and reviews the material; from 15:00 to 17:00 (adjusted for the Khenpo's schedule), Khenpo Tshul khrims Blo Grus[40] teaches in Tibetan; from 17:30 to 19:00, Khenmos or assistants repeat the morning teaching or handle inquiries; and from 19:00 to 21:30, they engage in individual study. They practice recitations for exams once every 3 to 4 days.

The exams for the Tibetan Khenmo degree were standardized in 2013. The exams had four parts, oral explanation (*'chad rgyugs*), recitation (*blo rgyugs*), a written exam (*yig rgyugs*), and a debate (*bgro gleng*)[41]. Each of the five major subjects of exoteric Buddhist study has four exams (Slob don khang 2017b). In 2013, there was a particular exam named *mtho rim gyi rgyugs*[42] for 200 Tibetan nuns who had high grades from different classes, to encourage them to study hard to get the top s50res and become Khenmos in the future. One hundred and six Tibetan nuns passed this exam and started four exams of the five major subjects of exoteric Buddhist study, then took two of the five (*gzhung po ti lnga*) exams every year, and these exams took three years to finish. Fifty-eight Tibetan nuns scored more than 90 and passed the four exams in 2016 but did not receive the Khenmo certification until 2018[43]. Fifty-six from this group[44] began teaching formal courses in February 2017. Thus, 104 Khenmos have been qualified and able to teach since 2017. Eighteen went to five other nunneries[45] to teach in 2017. Two Khenmos returned to Larung Gar, and sixteen continue teaching at those five nunneries. The five nunneries are Lab dgon in Yul shul, Rtse re bsam brtan chos gling in Gser shul Rdzong, Nyi 'od dgon in Chab mdo, and Btsun m'i dgon in Rta gsar. Eighty-eight stayed in Larung Gar teaching courses in 2018[46]. Twenty-seven nuns did not get the higher grades on the three-year exams in 2016, and since then, they continued to study the five major subjects of exoteric Buddhist study and must score at least 90. Twenty-one nuns passed the four exams of the five major subjects of exoteric Buddhist study in 2019 and got the Khenmo degree. Table 6 shows there were three different occasions for Khenmo certified after Khenpo Jigme Phuntsok.

**Table 6.** Numbers of certified Tibetan Khenmos by year after Khenpo Jigme Phuntsok, until 2019.

| The Year Exams Were Taken | The Year of Formal Certification of Khenmo Degree | Time | Number of Khenmos |
|---|---|---|---|
| 2009 | 2013 | Sixth | 28 |
| 2016 | 2018 | Seventh | 58 |
| 2016 | 2019 | Eighth | 21 |

There were two times (2013, 2018) when Khenmos were appointed, and 86 new Khenmo titles were earned and awarded after Khenpo Jigme Phuntsok passed away. There was another opportunity for 27 nuns to become Khenmos if they completed the exams, but only 21 earned the Khenmo title in 2019.

One significant development of the third period is the standardization of exams. Individual Khenmos gave oral explanation and recitation exams[47] to their students before 2012. After 2013, the Education Management of Larung Gar added two additional exams, writing and debate, and other Khenmos who did not teach the classes proctored the exams. The educational requirements include the frequency and subjects of the exams[48]. (1) The oral explanation exam (*'chad rgyugs*) is held three times a year. The examination intervals are 29 April to the end of the month, 29 July to the end of the month, and 29 August to the end of the month. (2) The recitation exam (*blo rgyugs*) is held twice a year. The exam occasions are 26 and 27 March and 21 and 22 November, respectively. The root text is recited on the first day, and the commentary text is recited on the second day. Students who cannot recite 50 poems cannot take the recitation test. Those who can recite more than 50 poems need to write 1 poem from memory; those who can memorize 100 poems need to write 2 poems from memory; those who can recite 110 poems need to write 9 poems from memory. (3) The written exam (*Dri rgyugs*) is held twice a year. There are two types of written examinations. One is the written examination that the class must take in June to prepare for the annual exam; the other is the annual exam held from 1 to 23 November. (4) Debate exams (*bgro gleng* and *rtsod rgyugs*) are held twice a year. The debate exams are held from 27 to 29 June, and from 16 to 23 October[49] (Slob don khang 2017b, p. 6). In addition to these prescribed exams, each class has six exams per month (*zla rgyugs*) from their own Khenmo and one annual exam (*lo 'jug gi rgyugs*) from the school of Larung Gar (Slob don khang 2017b). The debate exam has more detail and is explained in the document of education requirements. In the general debates, each class decides which students will participate in the examination by drawing lots/Mala. The debate exam at the end of the year requires the Epistemology classes to prepare and debate many times in advance in their own classes. The monthly debate table for each class needs to be planned in advance, which includes the dress, the number of participants in the debate, and the time and content of the debate. If there are students who cannot participate in the debate, they need to explain and cancel it with the manager of the debate exam. The head of each class has no right to cancel (Slob don khang 2017b, pp. 10, 12, 13). There is no requirement that "the Khenmo student must debate her instructors by following the classical style" (Liang and Taylor 2020, vol. 27, p. 249) for Tibetan nuns. In fact, two Tibetan terms are used in the regulations: One is debate (*rtsod rgyugs*), and the other is discussion (*bgro gleng*). It is mentioned in the regulations that the head of debate in each class should notify others about the content of the debate early, and if there is no debate at 17:30–19:00 for the day, other discussions need to be arranged. A Khenmo should now participate in the debate to arrange the format of the agenda (Slob don khang 2017b, p. 3). The topics and questions of the debate exam are arranged by the Education Management according to the main questions submitted by each class based on the teaching content (Slob don khang 2017b, p. 5).

The main subjects of the teachings and exams change every year. Compared with those of 2016, there were three grades in 2017, namely, beginner, intermediate, and advanced, which further divided students into different classes[50]. Different grades cover specific main

and reference content and exams, which follows the nuns' study subjects and is determined by their performance levels. For example, the Abhidharma exam (Slob don khang 2017a) in Table 7 is divided into three ranks and four different teaching contents and exams as required by the Education Management of Larung Gar. We can discern from Table 7 the three levels of **the annual and monthly oral explanation exams**: (1) The advanced level takes an exam on Abhidharma and includes the subtleties and explanation of the doctrine of the root text of Abhidharma, continuing to the end of the text. It occurs monthly and in April, July, and August, as required by Education Management of Larung Gar. (2) The intermediate level takes an oral explanation exam for Abhidharma and includes the subtleties and explanation of the doctrine of the root text of Abhidharma, looking at the chapters regarding the path, meditation, and perception of sentient beings. It occurs monthly and at the other three times noted above. (3) The beginner level takes an exam for Abhidharma that merely merely expands the doctrine from subtle and proliferating through the chapter on the world of sentient beings, again monthly and the other three times as noted above. **The written exam** is on Mipham's commentary of the Abhidharma, includes the subtle and explanation, analyzing Mipham's commentary for the chapter regarding the path, and the sentient beings, for all three levels twice a year. **The recitation exam** includes the root text of Abhidharma and the summary text of *Gateway to Knowledge* by Mipham (*Mdzod rtsa ba dang mkhas' jug sdom byang*) for all three levels, the advanced, intermediate, and beginner, and it occurs twice a year. The Education Management of Larung Gar requires the main content of Abhidharma as well as the reference, supplementary teaching of Abhidharma, which is about mind training and the principles of tantra.

**Table 7.** Example of the main subjects of teaching and exams in 2017[51].

| Course (*'dzin grwa*) | Grades[52] | The Main Content of Teaching and the Range of Annual and Monthly Oral Exams[53] | Range of the Written Exam[54] | Reference for Teaching[55] | Recitation Exam[56] |
|---|---|---|---|---|---|
| Abhidharma (*mngon pa*) | Advanced (*rab*) | From the subtles and explanation of the root text to the end[57] | From the subtles and explanation of Mepham's commentary and the chapters of path, and the sentient being[58] | Regarding mind training and the instruction of tantra[59] | The root text of Abhidharma and the summary text of Gateway to Knowledge by Mipham[60] |
| | Intermediate (*'bring po*) | From the subtles and explanation of the root text to the chapters of path, meditation, and perception of sentient being[61] | | | |
| | Beginner (*tha ma*) | From the subtles and explanation of the doctrine to the chapter of world of sentient being[62] | | | |

Changes occurred after July 2017, including the number of students in a class being limited to forty[63]. Another change was that the length of time needed for graduation was set at 15 years to finish the Culture[64] and Philosophy of Buddhism[65] courses, and 30 years

for those who continue to study the Tantra Section (*rgyud sde*) and the oral instructions (*man ngag*)[66]. The age and level of students studying in the classes for the five major subjects of exoteric Buddhist study have been clearly stipulated in the document of educational requirements. Nuns aged 15 to 30, and nuns who have passed the grammar exam and can correct typos, can enter the classes (Slob don khang 2017b, pp. 7–8). Table 8 depicts the number of classes for Tibetan nuns, based on the number of Khanmos who taught nuns of the different levels after 2017.

**Table 8.** The numbers of Khenmos who taught courses in 2018[67].

| Course | Number of Khenmos | Course | Number of Khenmos |
|---|---|---|---|
| Culture | 8 (grammar) 1 (second grade of poetry) | Prajnaparamita | 9 (beginner) |
| Vinaya | 3 (beginner),3 (advanced) | Madhyamika | 4 (beginner), 2 (advanced) |
| Abhidharma | 3 (beginner),1 (advanced) | Epistemology | 14 (beginner), 21 (advanced) |
| Mind Training (*Blo spyong*) | 2 (first grade), 4 (second grade), 3 (third grade), 9 (fourth grade), | Tantra Section | 1 (beginner), 1 (oral instructions, *man ngag* in Tibetan) |

There were 89 courses taught by Khenmos, but there were about 38 classrooms in the nuns' education building. Two or three classes take turns using each classroom. A class meets for one hour and forty minutes, and each course has at least two meetings a day. The Khenmos, who use the classrooms, determined the schedule for classes.

The school also started to have two regular semesters after 2017, as all secular schools have a Spring Semester for 4 months and a Fall Semester for 4 months, and Tibetan nuns have about a one-month break between the two semesters. There were some courses not taught by Tibetan Khenmos in 2018. Table 9 shows what courses were not taught by Khenmos after 2017.

**Table 9.** Some courses **not taught** by Khenmos in 2018.

| Course | Number of Classes | Course | Number of Classes |
|---|---|---|---|
| Tibetan Medicine | 2 | Arts and Crafts (*Bzo rig*) | 1 (first grade), 1 (second grade) |
| English | 1 (second grade) | Astronomy (*Skar rtsis*) | 1 (first grade), 1 (second grade) |
| Chinese | 1 (second grade) | | |

There are a few courses for Tibetan nuns that are not taught by Khenmos. The Arts and Crafts (*Bzo rig*), Astronomy (*Skar rtsis*), and Tibetan Medicine courses are taught by Khenpos. Classes in Chinese and English literature for Tibetan nuns (*yig rigs*) by a Chinese nun began in 2017. When doing field research and accompanied by a graduate student at Larung Gar in July of 2018, we observed a class outside the nuns' education building studying Chinese. A Chinese nun was teaching Chinese to Tibetan nuns.

After 2017, societal changes brought many challenges for the nuns in Larung Gar. Khenmos worked hard, and nuns diligently studied and practiced, and added a short meditation at the end of each class.[68] The administration added fire stairs in all directions and a road around the perimeter. Unfortunately, the latter increased traffic and noise and affected privacy. Larung Gar was rebuilding some new facilities in 2018, and when there,

the author witnessed Khenmos and nuns facing new situations and opportunities every day.

The changes included the number of nuns who can stay in Larung Gar, how many students in one classroom, how nuns can settle in and learn well, and how the area can become a "paradise." The community plans to make Larung Gar a park type of area, with green vegetation and flowers.

## 4. Conclusions

When delivering lectures[69], the author noted down attendees' questions and used them to guide the direction and development of fieldwork. For example, people frequently asked, "What is a Tibetan nun's daily life like, and what is their education like?" This article is based on Tibetan nuns and their lives, and on information from interviews in Tibetan in Larung Gar.

During the 1990s and early 2000s, nuns at Larung Gar studied *The Words of My Perfect Teacher* (*kun bzang la ma' I zhan klong*) in the *Jomos'* large hall and learned the teachings from Khenpo Jigme Puntshok. The primary role of Khenmos at that time was to assist, facilitate, and review the teaching of all Khenpos. In 2003 when the author was at the Larung Buddhist Institute, a Khenpo taught from *The Words of My Perfect Teacher*. Jetsuma Mumtso sat behind the *Lama* in the *Jomos'* large hall[70].

Formal classes in Buddhist philosophy were offered by Khenmos beginning in 2005 after the encouragement of Khenpo Tshul khrims Blo Grus. The progress of the new education system is apparent. It has progressed from the study of *The Words of My Perfect Teacher* in the *Jomos'* large hall to studying Buddhist philosophy and classes taught by Khenpos to Khenmos. Thus, the highest level of nuns' education was realized in 2005 when Khenmos were recognized as teachers themselves and began to teach a formal course titled "The five major subjects of exoteric Buddhist study" to *Jomos*. There have been systematic developments in their exams and curriculum.

The establishment of the new curriculum for nuns represents a radical improvement in nuns' education. The improved curriculum is revolutionary and a significant improvement for the Tibetan Buddhist education system. It is a significant elevation of the nuns' status. Khenmos, female monastic professors, can teach at least eight formal sections: Culture (*rig gnas*), Vinaya (*'dul ba*), Abhidharma (Abhidharmakosa, *Mngonpar chos' mdzod*), Prajnaparamita (*phar phyin*), Madhyamika (*dbu ma*), Epistemology (*tshad ma*), and Tantra (*rgyud sde*). The nuns have been using the same textbooks as the monks.

There have been many notable changes in Tibetan Buddhist culture over the years, ranging from the trends of vegetarianism and non-violence to the popularity of *Lamas* who teach the Dharma globally. However, perhaps the most significant development is a rigorous education system for nuns and the ability to become Khenmos. Tibetan Buddhist nuns are making history by earning the title "Khenmo". This title can be earned after many years of dedicated study, rigorous scholarship and practice, and higher monastery positions. The new education system has not only improved the status of nuns but has also given them new life and a very bright future.

The educational opportunities for Tibetan women and nuns have come a long way over the last 30 years. Following further developments and refinements in this area, there are also some differences, such as in texts,[71] the manner of teaching[72], the courses[73], and the challenges between Tibetan nuns and Han Chinese nuns in Larung Gar.

In the same online forum and conversation as referenced above, it was noted that similar changes may be happening at other monasteries. The author strongly encouraged colleagues to consider further researching these other monasteries and the changes happening there. To see if these changes and opportunities are spreading, how far, and in what forms, would be an excellent area for future research. The author is already engaged in research regarding some of these other nunneries.

An area for future research that naturally arises from educational changes and opportunities, which are extensively chronicled in this article, is that of women's rights and

activism, particularly related to changes found in the monastic setting of Larung Gar. Similar changes also seem to be occurring in the general culture, and the author is currently working on an article regarding this topic.

As of the time of writing, during the COVID-19 pandemic no one in Larung Gar caught the virus as the local government did not allow people to go outside after 27 April 2020. The Management Committee of Larung Gar issued public statements about not being open to visitors[74]. The pandemic did not infect the institute at Larung Gar, but the six public rituals became five rituals[75] and without any visitors. The school of nuns delayed the Spring Semester because of the pandemic. Tibetan nuns started teaching on 22 April 2020 and continued until early July, and then the Autumn Semester ran from the middle of July to about 20 January 2021 during the pandemic. It would be very interesting to learn whether COVID-19 has reached the nunneries, the current status of the virus, and effects it might have.

The higher education issue for Tibetan Buddhist nuns relates to improving the quality of life and the educational status of Tibetan women. It also impacts women's overall rights and equality and the growth of feminist ideas in Tibetan culture. Furthermore, it relates to international concerns about Buddhist women's issues, how Tibetan nuns take the bhikkhuni vows, and the meaning of feminism for Tibetan women. Tibetan nuns have been working very hard to increase their knowledge and rank in monasteries. There are still many challenges that Tibetan nuns face, such as training other high-quality Tibetan nuns, the question of how to have real equality with *Lamas* in monasteries and secular settings, and the next steps regarding social changes in modern Chinese society. Nevertheless, now many Khenmos have recognition in Tibetan monasteries, especially in Larung Gar. They are diligently working on elevating Tibetan nuns' education and talking about Tibetan women's status and rights ((Baimacuo) ((Baimacuo)) 2020), thus impacting the future of many other Tibetan monasteries and Tibetan women.

**Funding:** This article is part of the research project "Zhulao zhonghua minzu gongtongti yishi yanjiu: Zangchuan fojiao zhongguohua de jianxingzhe Maoergai Sangmudan" (21GTYBC04) supported by "Zhulao zhonghua minzu gongtongti yishi yanjiu zhongxin of Southwest Minzu University", and "The International Impact of Tibetan Buddhism: A Case Studies of Nyingma Lineage (16BZJ014)".

**Conflicts of Interest:** The author declares no conflict of interest.

## Notes

1    Larung Gar, *Lba rung sgar* in Tibetan, *Seda Larong Wuming foxueyuan* in Chinese, is located in Sertar county of the Kham Tibetan area. It was established by Khenpo Jigme Phuntsok (*Jing mei peng cuo fa wang* in Chinese) in 1980. This was the sacred practice place of *'dul 'byum Rinpoche*. Larung Gar is the largest Tibetan Buddhist monastery and Dharma institution in China now. There are many thousands of monks and nuns who stay there and study Tibetan Buddhism. Larung Gar is the most important and influential Tibetan Buddhist monastery and Dharma institution in the contemporary Tibetan area of China.

2    "Chinese-Tibetan Buddhists" means Tibetan Buddhist disciples who are Han Chinese.

3    Lama is *bla ma* in Tibetan, which means a Tibetan Buddhist master.

4    Khenmo is *mkhanmo* in Tibetan, a female Buddhist teacher who teaches Tibetan culture and Buddhism in a nunnery in the Tibetan area after finishing the traditional monastic scholastic study and exams. The Tibetan word *mkhanmo* is written as Khenmo in this article because the spelling Khenmo is accepted by Western Tibetan scholars.

5    Khenpo is *mkhanpo* in Tibetan and is a traditional title and esteemed status for Tibetan Buddhist monks who have finished their studies (ranging from 13 to 20 years (Hongxue 2012, p. 131)), passed the tests and debates, and completed the lengthy retreat in a Tibetan monastery. They are then able to give teachings to the monks and nuns in monasteries. The Tibetan word *Mkhanpo* is written as Khenpo as this spelling is accepted by Western scholars.

6    The fieldwork at Yachen started in July 2006, and I returned and continued the fieldwork in July 2007, May 2008, July 2010, and July 2011.

7    The five major subjects of exoteric Buddhist study are called *Gzhung po ti lnga, Gzhung bka' pod lnga, Gzhung pod lnga,* or *Gzhung chen pod lnga* in the Tibetan language, encompass the entire Dharma teachings of Buddhism, and are divided into main five subjects of Tibetan Buddhism as taught in monasteries. The five subjects are Prajnaparamita (*phar phyin*), Madhyamaka (*dbu ma*), Epistemology (*tshad ma*), Abhidharma (*mngon par chos mdzod*), and Vinaya (*'dul ba*). Each of the subjects follows the sutras and

corresponding commentaries to explain about wisdom and the paths of the mind (*sa lam*), the middle way, valid cognition and logic, and the knowledge of physical and mental constituents of beings.

8  *Jomo* is the name for a nun in the Tibetan language. A Tibetan nun, or *Jomo*, has taken the ten precepts, or the basic Buddhist vows called *Dge tshul ma* in Tibetan.

9  Zungchu, Song Pan, is located in northeastern Aba Tibetan area. Fieldwork at Bya'du was in July of 1995, 1996, 2002, 2008, and 2020.

10  Lho dzong nunnery, Lho rdzong jo mo dgon pa in Tibetan, Nan Zong Ni Gu Si in Chinese, is located in Kan la township, Jianzha county, Huangnan Prefecture. Fieldwork at Lho dzong was in July 2000.

11  Ani Tshankhung nunnery, *A ni tshang khung jo mo dgon* in Tibetan language, *A ni cang kong si* in Chinese, is located in the city of Lhasa. Fieldwork at Ani tshan khung was in July 2000.

12  Yachen Gar, also call Yachen or Ya chen o rgyan bsam gdan gling in Tibetan, is located in Palyul county of the Kham Tibetan area. Acho Rinpoche officially set up the Yachen Monastery in 1985. The monastery is the manifestation of the Palace of Infinite Light of Padmasambhava (*Bzangs mdo dpal ri*).

13  Khenpo Jigme Phuntsok, *mkhanpo 'jigs med phun tshogs 'byung gnas* in Tibetan language, 1933–January 7, 2004, was an incarnation of Terton Sogyal. He was one of the most important Khenpos for Tibetan Buddhism in contemporary China after the Cultural Revolution.

14  Since 2010, one hundred questionnaires regarding education were answered by students from different courses, and there were more than sixty interviews about Khenmos' degrees, their teaching methods, and their life stories.

15  High Lama means a Tibetan Buddhist master who has the signs of accomplishment and is respected by local people, and has completed the entire, strict Vajrayana teachings and retreat.

16  *Dge bshes* is degree level in the Tibetan Buddhism education system, mainly used in the *Dgelugs*' Buddhist academy.

17  Khenpo Tshul khrims Blo Grus is the Tibetan for Khenpo Tsultrim Lodrö. He was born in 1962 in Drag'go (Ch: Luhuo) County in the Kham Tibetan area. He is one of the main students of Khenpo Jigme Phuntsok, and he is one of the most important Khenpos for contemporary Tibetan Buddhism in China.

18  Jetsuma Mumtso is called mkha' 'gro ma mu med ye shes mtsho gtso/mo in the Tibetan language. She is the niece of Khenpo Jigme Phuntsok, and became the head abbot/abbess in Larung Gar after Khenpo Jigme Phuntsok passed away.

19  Most of the names in this article are not their original names. They may have just one sound mark in their name. Khenmo *Chos dbang* is about 45 years old and has lived in Larung Gar for about 30 years. We have been friends for more than seven years.

20  Use of the term Lama, in this paper, means a Tibetan Buddhist Master, and refers to Khenpo Jigme Phuntsok in Larrung Gar. There are many other Lamas who give teachings in Larung Gar and Tibetan society, but when people talk about the Lama in Larung Gar, that usually means their main and important Lama who is Khenpo Jigme Phuntsok in Larung Gar.

21  I interviewed Khenmos in 2010, 2013, 2014, and 2016. I was told there were four *Jomos* receiving the title of Khenmo for the first time. I interviewed a Khenmo who was the head of Si Guan Hui three years ago and she reported the number and the year of Khenmos. This number came from her interview in June 2018.

22  Lama means a master, and this refers to Khenpo Jigme Phuntsok.

23  I interviewed one Khenmo in June 2018 at Larung Gar. The Khenmo was titled by Khenpo Jigme Puntshok in 1997.

24  According to a few interviews with Khenmo Chos dbang, she confirmed the date with another Khenmo. The first cohort graduated in 1997 (Liang and Taylor 2020). Han Chinese nuns started to learn the five major subjects of exoteric Buddhist Study (*gzhung po ti lnga*) in about 2003. The first Han Chinese Khenmo was awarded in 1991.

25  Lama (*bla ma*) is master or teacher in Tibetan. Here, Lamas give practitioners guidance, Dharma teachings, transmissions, and empowerments as Jetsuma Mumtso does.

26  Si Guan Hui is the Management Committee in Chinese, and three or four monks or nuns of a monastery with the Bureau of Religious Affairs cooperate to keep religious activities on track. The head of Si Guan Hui for nuns changes frequently at Larung Gar.

27  Khenmo *Rin 'dzin* in Tibetan language. She is about 30 years old and has lived in Larung Gar for about 14 years. We have been friends for over five years.

28  The three disciplines were required by Khenpo Jigme Phuntsok in Larung Gar. All monks, nuns, and lay practitioners (*bla ser*) who stay at Larung Gar must observe and keep the three disciplines, in Tibetan "*Thug mthun dang khrims gtsang po dang sbang ba'i gtsang pa dang dpe slobs dang sgom rtson pa*", and these are group cohesion and solidarity, the keeping of disciplines and abstaining, and pursuing diligent study and meditation. Discussing the various disciplines in more detail is a subject for a future article.

29  Assistants are nuns who can help with clarifying the teachings of the Khenmos, and they tutor the nuns to help them study and understand what the Khenmos taught for that day.

30  There was one Chinese nun who earned the title of Khenmo in 1991. Her name is Zhao wu. She became a nun at Emei Monastery in 1988 and came to Larung Gar in February of 1990. She started to study Buddhism and the preliminary practice of Vajrayana. Han Chinese nuns who come from Inner China and different districts have been studying Tibetan Buddhism in Larung Gar since the end of the 1980s. They mainly learn from Khenpo Sodargye, who translated Khenpo Jigme Phuntsok's teachings into Chinese

31  before 2004. Khenpo Sodargye started to teach the five major subjects of exoteric Buddhist study (*gzhung po ti lnga*) in Chinese in around 2003 for Han Chinese nuns.

31  *Blo spyong* means mind training, or self-cultivation. The main book on mind training is *The Seven Points of Mind Training*, by Atisha, of the Kadampa tradition.

32  The master refers to Khenpo Sodargye for Chinese nuns since Khenpo Jigme Phuntsok passed away. This interview was in 2013.

33  The number came from interviews in 2010 and 2017. According to the interview in 2010, there were thirty-three Khenmos who could teach the five major subjects of exoteric Buddhist study (*gzhung po ti lnga*).

34  Khenmo Yon tan, also called Thub bstan rig byed lha mo, was 37 years old when I interviewed her in 2013. She came from Gyegu of Yushu and became a nun in February 2003. She went on to study with Khenpo Jigme Phuntsok in March of 2003 and got her Sramanera (*dge tshul·*) from *Khenpo Ri grol*. She wrote eight books of commentaries on the five major subjects of exoteric Buddhist study (*gzhung po ti lnga*) from the 53-volume Ḍākinīs' Great Dharma Treasury, edited and published by the Larung Ārya Tāré Book Association Editorial Office in 2017.

35  There were many Khenmo classes held in tents when I was there in 2010.

36  I was given the educational requirements when I did fieldwork there in 2016.

37  *Skal bzang mthso mo* (Kal zang tso mo), *Rnyed dka' ba'i nga tsho'i dus tshod* (our Rare time), *Gangkar Lhamo* 2013: 73.

38  The numbers of Khenmos who taught the main formal courses came from two Khenmos I interviewed in July 2017. However, a Khenmo from Education Management reported there were forty-eight Khenmos teaching courses. The numbers in the table are the minimum number of Khenmos who were teaching the main courses in 2017.

39  The schedule for morning rising of the Chinese nuns is different than that of the Tibetan nuns. Many Han Chinese nuns in Larung Gar get up at 04:00 for meditation (Liang and Taylor 2020), while Tibetan nuns usually get up at 05:00 to 05:30. The reason for this difference seems to be that a lot of Han Chinese nuns keep their Chinese Buddhist tradition of earlier rising while in Larung Gar. Another difference between the two is that during exam times, some Tibetan nuns spend the entire night in the classroom, just sleeping a few hours and studying most of the night and early morning in preparation for the exams. The difference is also due to the academic semester; Han Chinese nuns have one to three months of winter break and resume in March or April (Liang and Taylor 2020), but Tibetan nuns don't have a long winter break. They had a few days' break after winter and summer exams until 2017. The schedule of Tibetan Khenmos and assistants is adjusted to the schedule of Khenpo Tshul khrims Blo Grus's teaching in the afternoon.

40  Khenpo Tshul khrims Blo Grus teaches in Tibetan year-round, except for October through most of January. Tibetan nuns can listen to his live teachings in their big hall or their residences. So, there is no winter break for Tibetan nuns but there is for Chinese nuns (Liang and Taylor 2020).

41  Tibetan nuns started to learn and practice debate (*bgro gleng*) after 2014. When I was in Larung Gar in 2013, I discussed with the manager of Tibetan Education Management about why Tibetan nuns don't have as much debating as the monks do. She told me that it was because some people think that Tibetan nuns don't have this tradition in the Tibetan Buddhist history, but she then went on to add, "We have a little debate and discussion in our study." The Tibetan Education Management added a debate (*dgro gleng*) exam and subject for Tibetan Khenmos into their required studies at Larung Gar in 2017. The requirement indicates that the debate exam (*rtsod rgyugs*) will be held twice a year; one exam is June 26 to 28, and the other is October 16 to 23.

42  *Mtho rim gyi rgyugs* meaning "the exam of high level". This exam is for the election of Khenmo candidates in 2013.

43  In July of 2018, two new Khenmos told me they will get the certification this fall from Jetsuma Mumtso, Khenpo Tshul khrims Blo Grus and other Khenpos. During the three-year exams, they passed each of the five major subjects of exoteric Buddhist study (*gzhung po ti lnga*) in a different year.

44  The number came from a Khenmo in Education Management.

45  The five other nunneries are the nunneries who invited Larung Gar Khenmos to give teaching to their nuns. The names of the five nunneries follow in the article.

46  The numbers came from the Tibetan Khenmo in Education Management whom I interviewed January 2018.

47  For Tibetan nuns, the oral and recitation exams are separate from each other. All students in a class must have passed the recitation exam of the root text or what the Khenmo asks. The author experienced some Tibetan nuns who visited the Khenmo's room for recitation exams. Han Chinese nuns' oral exam usually consists of recitation (Liang and Taylor 2020), so there are three exams (Liang and Taylor 2020) for Han Chinese nuns, and Tibetan nuns have four exams, which are explained in the following footnotes and in Table 7.

48  The details of the education requirements are in the *Bla rung padma mkha' 'gro'i 'du gling gi 'dzin grwa so so'i sbyang gzhi dang blo len gyi skor* and *Slob gnye gyed stangs skor sogs gros chod byung ba'i rim ba* for 2017. The exam system for Han Chinese nuns has both the oral and writing exams in Chinese. They started administering the exams much earlier than those for Tibetan nuns, due to the Han Chinese nuns' classroom being built and formal course available much earlier than those provided for Tibetans.

49  The four exam dates come from the author's interview with Tibetan Khenmos and the document of education requirements. All the dates are from the Tibetan calendar. Following the education requirements, Tibetan nuns have to take the four exams and do what the rules require.

50 I got the subjects of teachings and exams for 2017 in January 2018. A Tibetan Khenmo from Education Management explained that they are following new material this year.

51 The range of exams changes every year. This information comes from the study and examination documents of each class in *Padma mkha' 'gro'i 'du gling* at Larung Gar in 2017.

52 *So so' i blo rim kyi dbye ba* in Tibetan. There are three grades, which are advanced (*rab*), intermediate (*'bring po*), and beginner (*Tha ma*).

53 *Stsa ba'i 'khrid bya dang lo mjug rgyugs kyi gzhung dang zla rgyugs kyi gzhung* in Tibetan.

54 *Dri rgyugs kyi gzhung* in Tibetan.

55 The reference for teaching is *Zur 'khrid sgor* in Tibetan, and is about supplementary teaching of the root text.

56 The recitation exam is *blo rgyugs* in Tibetan, and the contents of the recitation exam are indicated in the document.

57 *Mdzo rtsa'i phra rgyas bstan pa nas 'jug bar* in Tibetan.

58 *Mipham mchan 'gres gyi phra rgyas dang lam dang gang zag bstan pa* in Tibetan.

59 *Blo spyong skor dang sngags zhal gdams skor* in Tibetan

60 *Mdzod rtsa ba dang mkhas 'jug sdom byang* in Tibetan.

61 *Mdzod rts'i phra rgyas bstan pa nas lam dang gang zag bstan pa'i bsgom dang mthong la rab tu phye yan* in Tibetan.

62 *Phra rgyas kho na srid pa'i zag yan* in Tibetan.

63 The number of nuns who can stay in Larung Gar is limited to 3500 since July of 2017.

64 Culture study, *Rigs gnas* in Tibetan, includes Tibetan grammar, literature, and poetics, Chinese, and English, since 2017. The Education Management added Chinese and English classes into Culture study.

65 Six years of the Philosophy of Buddhism requires each of the five major subjects of exoteric Buddhist study (*Gzhung po ti lnga*) for at least two years. Culture (*rig gnas*) includes one year of Tibetan grammar, and at least two years for the study of Literature in Tibetan, English, and Chinese. Meanwhile, History, Computers, and Mind Training (*blo sbyong*) require three to five years.

66 The oral instruction (man ngag) is the essential instructions/method, to help the practitioner successfully practice Vajrayana and Buddhism. This is given by the master (*Lama*) whom you follow and study with.The main oral instructions in Larung Gar are the linage of Longchenpa.

67 The number came from a Tibetan Khenmo in Education Management in August of 2020.

68 When I was in Larung Gar in 2018, I attended a few Khenmos' teachings, and Khenpo Tsultrim Lodrö's streaming teaching with Larung Ārya Tārē editors. All the classes concluded with a short meditation. This is a new event in the classroom. Previously, I did not have this experience during my fieldwork.

69 The lecture "Contemporary Education of Tibetan Buddhist Nuns in Kham, Eastern Tibetan Area" was presented at the Divinity School of Harvard University in 2011, and at the University of San Diego and Naropa University in 2011. The lecture "The Lives of Nuns at Larung Gar" was presented at the Himalayan Studies Conference at the University of Colorado in 2017.

70 My friends of nuns told me, "we are going to the teaching of Mkha 'gro ma mu mtsho." I went with them, but I was so surprised and saddened that it was actually a male teacher (Khenpo) giving teaching! Jetsuma Mumtso (mkha' 'gro ma mu med ye shes mtsho gtso or mkha' 'gro ma mu mtsho) just sat behind the male teacher (Khenpo).

71 Chinese nuns in Larung Gar have been using Buddhist texts in Chinese about the five major subjects of exoteric Buddhist study, which were translated by Khenpo Sodargye and Kenpo Yeshes Phantsho, and some reference Chinese Buddhist texts.

72 There are two main teaching methods for Han Chinese nuns: The first is Khenpo Sodargye (*mkhan po bsod dar rgyas* in Tibetan) teaching every night. Every time that I was there, he taught around the middle of the night in the Chinese nuns' hall; the second is Han Chinese Khenmos teaching in Chinese nuns' classrooms, which are in a building inside their hall. When I was there in 2003, there were Han Chinese assistants teaching in their own rooms.

73 Han Chinese nuns formally started classes on the five major subjects of exoteric Buddhist study (*Gzhung bka' pod sang*) of Tibetan Buddhism in Chinese, taught by Khenpo Sodargye, in 2003. When I went to classes taught by Khenpo Sodargye and his Han Chinese assistant in 2003, they taught the Abhidharma.

74 The Document of Larung Monastery Management Committee of Ganzi Tibetan Autonomous Prefecture: Supervision of Ganzi Larung [2020] Number 38 (Chinese: Gan zi zang zu zi zhi zhou la rong si guan wei yuan hui wenjian: Gan la guan [2020] 38 hao), the Office of La Rong Monastery Management Committee (Chinese: La rong si guan li wei yuan hui ban gong shi), 27 April 2020 (Chinese: 2020 nian 4 yue 27 ri).

75 The five rituals are the Assembly of 9 deities (Gar dbang lha dgu), Vajrasattva (Rdo rje sems pa), Samantabhadra (Kun du bzang po), Kshitigarbha (Sa'i snying po), and Amitabha ('od dpag med) and 'dus mchod. The ritual of long life (tshe sgrub) was the one that was cancelled because of the pandemic in 2020.

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
