# Peer review of "How Tibetan Nuns Become Khenmos: The History and Evolution of the Khenmo Degree for Tibetan Nuns"

_religions, doi:10.3390/rel12121051_

Round 1
Reviewer 1 Report
The author introduces some novel information, based on two Tibetan monastic communities, about monastic education among Tibetan Buddhist nuns in contemporary Tibet. In addition, the relevant statistics provided by the author are very useful for understanding the actual changes occurring for the status of Tibetan nuns in Tibetan monasteries. However, these valuable points are undermined by typos and discrepancies in names and terms the author uses throughout the article. Because this article aims at introducing new information about, rather than analysis of, communities that are not easily accessible to ordinary people or researchers, these kinds of errors can affect the credibility of the information the author provides. I will provide a list of the typos and inconsistent uses of terms toward the end of my review. This article is not yet ready for publication and needs substantial revision.
The author claims that the paper is based on field research done in two important nunneries (Larung and Yachen, though it is not clear to me that Larung and Yachen can be called “nunneries”). However, throughout the paper, the data used by the author appear to come exclusively from field work done in Larung Gar. If this is not correct, the author needs to clearly indicate which statistics/data are drawn from Yachen Gar. In relation to this, and perhaps more crucially, the author seems to treat the education programs for the nuns in Yachen and Larung as very similar or identical; but since the author also states that these two communities are “very different” (66) and has only given data for Larung, the author should clearly indicate if and why the educational programs for the nuns in these two communities can be treated in the same manner.
Overall, the author’s argument is a little simplistic. After providing the increased number of nuns in Larung Gar with the Khenmo degree, the author states that this has “altered the Tibetan Buddhist education system and shaken traditional Tibetan society” (142-143), but the author doesn’t tell us what changes are actually occurring. I am doubtful that it is reasonable to claim that traditional Tibetan society has been “shaken” based on changes in Larung Gar. If this is really so, the author should provide a more thorough analysis with supporting data. In addition, the author bases her/his claim about the nuns’ improved educational opportunities and their improved status on the increased number of classes and formal/diversified subjects offered to the nuns. These numbers are very important evidence, but this is the only proof that the author provides (repeatedly) for her/his argument. The discussion would be much richer if the author included how the ordinary nuns feel about these changes. Do they prefer to take Khenmo’s classes over Khenpos’? Are the nuns satisfied with, or not satisfied with, the Khenmo’s classes overall, and why? Since the author did extensive field work in the community, it is reasonable to expect the inclusion of data regarding the ordinary nuns’ opinions.
I am glad to see that the author actually visited Larung after the massive destruction and evacuation of the nuns by the Chinese government since 2016. It has been reported that both Larung and Yachen have undergone significant reductions in terms of the numbers of practitioners, visitors, and other public activities. However, the author never indicates what significant structural changes have taken place in Larung Gar and claims that, in any event, the nuns’ educational system has not been affected. If so, this is surprising given the scale of the reduction and reported changes, and the reasons why the educational system has not been affected should thus be addressed. In the lines 377-380, the author in fact mentions changes occurring in Larung Gar, but the information given is neither sufficient nor clear: “how many students in one classroom, how nuns can settle in and learn well, and how the area can become a ‘paradise’”. I am especially puzzled by the author’s use of “paradise” in this sentence (and the use of quotation marks). Is this meant as a sarcastic reading of the community as a “paradise”? The tone of the sentence doesn’t read as sarcastic, so I do not know why the quotation marks are necessary, the meaning of “paradise” should be made clear.
In addition, this article would benefit greatly from professional editing.
In the following, I provide some specific suggestions for revision.
Jomos vs. nuns: the author uses both “Jomos” and “nuns.” In footnote 4, the author states that “Jomo: nun with ten precepts or the basic Buddhist vows called Dge tshul ma in Tibetan.” Are “Jomos” and “nuns” used interchangeably throughout? Why is “Jomos” capitalized?
Lama: in footnote 14 the author explains this term for the first time in the following way: “Lama means a master, and this refers to Khenpo Jigme Phuntsok.” This definition asserts that only Khenpo Jigme Phuntsok is a Lama, which is obviously false. The author uses “Lama” and “Lamas” throughout the paper when not referring to Khenpo Jigme Phuntsok. This term should be properly defined when it first appears (for example, in line 21) and the author should carefully distinguish when she/he is using it to refer to a specific figure and when she/he is using it to refer generally to Tibetan masters.
(19-20) “There has been…in both men and women Chinese-Tibetan Buddhists”: What does the term “Chinese-Tibetan Buddhists” mean? Does it indicate ethnically mixed Buddhist disciples?
(27-30) The author begins by citing two specific articles about Yachen and raising several questions about the nuns and their education, but the remaining paper is exclusively about Larung. Thus, it is not clear if these questions are meant to be about the nuns in Yachen, in Larung, or both, or even about Tibet as a whole. As the author states, Larung and Yachen are different, but the argument being made is based on the assumption that these two communities are very similar in terms of the nuns’ education. The author should clearly indicate how this is so. Otherwise, I do not see why these two particular citations regarding Yachen are given in the introduction or what their connection is to the rest of the materials in the article.
(44-46) The author mentions that extensive fieldwork has been done in several places in Tibet. Citations for these works are required.
(49-52) The author says that many nuns come to the two “nunneries” (Yachen and Larung), and “with the innovation and support of Khenpo Jigme Phuntso, help to change the education system of Tibetan Buddhism…” This sentence reads as if Khenpo Jigme “Phuntso” (misspelling) also helped to change Yachen’s education system and the education system of Tibetan Buddhism in general. Is it true that Khenpo Jigme “Phuntso” helped change Yachen’s education system? Is it reasonable to claim that he helped to change the education system of Tibetan Buddhism as a whole? Have other Tibetan monasteries, for example, those in the Amdo and Ü-Tsang areas, been influenced by Khenpo’s initiatives on nuns’ education? Are these other monasteries starting to adopt similar educational reforms for the nuns? The author should provide evidence for this.
(283) Footnote 31 should be placed at the beginning of the paper.
(331) Table 6 is incorrectly cited.
(314) What is the meaning of “another occasion” of 27 Khenmo degrees? Why is this not included together in the number of Khenmo degrees awarded “after” Khenpo Jigme Phuntsok passed away.
Inconsistent spellings and clarification issues:
*Khenpo Jigme Phuntsok—sometimes spelled it as Khenpo Jigme “Phuntso”.
The author also uses “Lama Khenpo Jigme Phuntsok” (167)
*Yachen—sometimes given as just “Yache.”
*Larung gar—sometimes spelled as Larong gar
*High Lama (169) Why should “High” be capitalized? Are “High Lamas” different from high-ranking lamas?
* “educational requirements” (264) The author say in footnote 29 that the author (himself/herself) was given educational requirements. Was the author educated in Larung? Or does this mean that the author was informed of the educational requirements? What are these “educational requirements”?
*Throughout the paper, the author uses different spelling conventions for Tibetan and inconsistently uses the Wylie and phonetic formats. Sometimes, phonetic formats are stated first, followed by Wylie spellings with parentheses; at other times, no phonetic formats are provided and only Wylie spellings appear. The Wylie format is only readable for those who are trained in Tibetology or a related field. In my opinion, the author should always provide phonetic spellings as well. In any case, the author should be consistent in ordering Tibetan spellings.
Reviewer 2 Report
This paper has been clearly written by a Tibetan and needs a lot of English corrections, which I can't do because of not being a native English speaker.
However, also for the Tibetan (and maybe also Chinese, but I am not a specialist in this and just saw two mistakes), there is a need of: orthography correections (use Tibetan original for example and go through https://www.thlib.org/reference/transliteration/wyconverter.php-tool), the same for English translitteration). There is also a lack of consequency: i.ex. Larung Gar, sometimes Larong Gar - adapt everywhere the same writing.
I did a couple of corrections, see pdf file attached. As well as some suggestions in order to finetume the text itself.
Reading other authors' articles on similar/same subjects can also help for the exact writing.

Reviewer 3 Report
The article under review is based on thorough research on the development of the Tibetan Khenmo program at Larung Gar from the late 1980s to the present. It draws on fieldwork conducted at Larung Gar spanning approximately one decade, including extensive interviews with Larung Tibetan nuns. The article makes effective use of visual aids such as tables and charts to present quantitative data. The author's central argument -- that the development of the khenmo program has matured significantly over the past three decades -- and parsing of this developmental trajectory into three key stages, is well-supported by the analysis and evidence presented in the article. The author is to be commended on his/her/their long-term commitment to fieldwork among the Larung khenmos and the rich and valuable data gathered in the process.
My overriding concern, however, is that the subject and contents of this article overlap considerably with a recently published article on the Larung Khenmo program by Jue Liang and Andrew S. Taylor ['Tilling the Fields of Merit: The Institutionalization of Feminine Enlightenment in Tibetan's First Khenpo Program' (2020) Journal of Buddhist Ethics Volume 27. Liang and Taylor's article addresses, among other things, Khenpo Jigme Phuntsok's motivations for creating the Khenpo programme, the date of arrival of nuns at Larung Gar, their initial teaching arrangements, the dates of Khenmo degree conferrals, the criteria for the conferral of khenmo degrees, the khenmo curriculum and rough course syllabi, nuns' daily study routines, class sizes, examination requirements, semester arrangements and graduation schedules. It also discusses the Larung Tibetan khenmos' extensive publishing and editorial activity. While the present article goes into more detail on some of these aspects than Liang and Taylor’s (and, in places, presents findings that differ slightly from those of Liang and Taylor), the focus and content of both articles are essentially very similar.
There is no evidence of plagiarism on the part of the author of the present article - it would seem that he/she/they was simply unaware of Liang and Taylor's already-published piece on the same topic. Nevertheless, there is too much overlap between the present article and the article by Liang and Taylor for the present article to be published as it currently stands. I have indicated the overall merit of this article as 'low' chiefly for this reason – and not because the article, taken alone, is without merit.
It is my recommendation, accordingly, that the article be reconsidered for publication after undergoing a process of major revision. I would suggest that the author closely consult the article by Liang and Taylor to identify areas of overlap. The author should then figure out which dimensions of the present article can be expanded upon, reframed, and supplemented by additional data to create the basis for a more original contribution to scholarship on the Larung khenmo program. While Liang and Taylor’s article is comprehensive, it is not exhaustive, and in view of the present author’s long term fieldwork among the Tibetan nuns at Larung Gar, one would imagine he/she/they is well-positioned to highlight other dimensions of their conditions for study and practice.
The author’s periodization of the development of the khenmos’ program into three stages: 1) initial transformation; 2) reconfiguration and 3) improvement and refinement is a useful conceptual framework for analyzing the evolution of the program. A revised and reworked version of this article could expand beyond its current focus to examine the development of the scholastic and contemplative programs of the Tibetan jomos side by side. Indeed, notwithstanding that Larung Gar has produced an impressive number of Tibetan khenmos in the past few decades, most Tibetan nuns enrolled at Larung Gar don’t become khenmos, and it would be interesting to know how the khenmo program fits within a multi-generational female sangha with diverse levels of literacy, scholastic aptitude, and Dharmic aspirations. A very significant proportion of the Tibetan female sangha have been historically involved in the contemplative (drubdra) track of Buddhist training. Perhaps an article that examines the evolution of the scholastic and contemplative tracks within the Larung nunnery could be a fruitful trajectory along which the author could rework this article.
Whichever way the article is reworked, it is important that the author engages more with existing scholarship on Larung Gar and female monasticism in contemporary Tibetan societies to show how her work builds on and diverges from what has already been written. While I realise that the Larung Tibetan female sangha is still under-researched, the work of Liang and Taylor (2020) as well as that of Sarah Jacoby and Padma Tsho (2020, 2021) must be made part of the conversation. Holly Gayley’s introduction to the Voices from Larung Gar volume should also be engaged with, and Elizabeth McDougall’s work on female contemplative practice in Eastern Tibet also merits consulting as a counterpoint to the scholastic emphasis at Larung Gar. Yasmin Cho’s research on Tibetan nuns at Yachen Gar should also be mentioned. These are just a few suggestions.
Given that I do not feel the present article can be published as it currently stands due to its overlap with Liang and Taylor’s existing publication, I will not go into too much detail about matters of English language and style, except to say that significant work here is needed to bring the article up to a publishable standard. While, in general, the author’s overall presentation of his/her/their findings is coherent, there are many instances in which a lack of clarity of expression impacts the article’s readability. A revised version of this article would benefit from English copy-editing prior to submission.
As a rule, Chinese or Tibetan terms should be translated into English the first time they appear in the article, with the Chinese or Tibetan transcription included next to the English word. Thereafter it is acceptable just to use the English translation (without the Chinese or Tibetan transcription), but consistency in this regard is important. For example, one page one, line 32, the author mentions the ‘five major subjects of exoteric Buddhist study’ with Gzhung bka’ pod snga in parentheses. In subsequent uses, it is acceptable to simply refer to the ‘five major subjects of exoteric Buddhist study’ without writing Gzhung bka’ pod snga again.
If the author wishes to leave a non-English language term untranslated, it should be italicised. Again, consistency is key. One page 2, line 71, the terms rdzogs pa chen po and ‘Bshad grwa appear without their English counterparts. This needs to be corrected.
The names of Buddhist texts should appear in English, with the Tibetan or Sanskrit names included in either footnotes or parentheses. As per above, it is only necessary to include the Tibetan or Sanskrit the first time the text is mentioned in the article.
The author needs to decide which system of Tibetan transcription he/she/they wishes to use and remain consistent in this usage throughout the article. As the article presently stands, a mixture of systems is used with no clear underlying principle, which looks careless and sloppy.
As for the names of religious figures at Larung Gar, the author needs to be consistent in the way that he/she/they refers to them throughout the article. For example, Mumsto is referred to as Mentsho mkha’ droma on page 3, line 102, but as Jetsuma Mumtso on page 13, line 391 (with Mumtso’s full title in parantheses). Khenpo Tsultrim Lodro’s name (page 3, lines 94 and footnote 12) is incorrectly written in Wylie as khred tshul blo grub. It would make sense to include the Wylie version of Tibetan names in parantheses the first time they appear, and thereafter to use a phonetic-based transcription method. The names of Khenmos Chos dbang and Ren’ dzen, for example, would be better rendered phonetically so that readers who are unfamiliar with Wylie are able to know how their names are pronounced.
As for place names, the author should include both the Tibetan and Chinese, but stick to using one or the other in the main text. Generally, the author should revise the conventions for rendering Chinese words in pinyin, as errors occur throughout the main text and footnotes. One page 4, line 130, for example, Luhuo County is written as Luhe County.
For some reason, the author renders ‘Yachen Gar’ as ‘Yache Gar’ throughout the article. It should be written as Yachen Gar.
On page 2 line 71 Larung Gar is mistakenly referred to as ‘The Larung Monastery.’ In footnote one on page one, the Chinese name for Larung Gar is written in pinyin as ‘La rong fo xue yuan.’ The correct official name is ‘Seda Larong Wuming Foxueyuan.’
Effort should be made to improve the quality of the footnotes. In footnote two on page one, ‘Khenmo’ is defined as a ‘female Buddhist teacher who teaches Tibetan culture and Buddhism in a nunnery in the Tibetan area.’ No mention is made of the scholastic qualifications that khenmos must possess or the connection between the title ‘khenmo’ and its male counterpart ‘khenpo.’ (In fact, the definition of khenmo should ideally appear in the main text and not be submerged in a footnote). In footnote three on page one, the five major subjects of exoteric study are said to ‘encompass the entire Dharma teachings of Buddhism’ when the author has already made it clear that these subjects do not encompass the teachings of Mahayana Buddhism’s tantric vehicle. There are many other examples where greater care, attention or accuracy is needed with respect to the information included in the footnotes.
Round 2
Reviewer 2 Report
The article has been considerably improved in terms of English and also the presentation of Tibetan terms. However, there is still a good deal of proofreading to be done on the English (there have been considerable additions after the first version) and the presentation of Tibetan terms needs to be standardized (e.g. Wylie Tibetan: in italics when usual words and without italics for Names such as towns, regions, personal names, etc.) Also when citing personal names, it is better to give first the Western reading and then in brackets the Tibetan Wylie (and never in italics).
There is a new long passage (p.13-15), which details the new study program of Tibetan nuns. However, I think this passage should be re-written in a way that it reads easier (too many Tibetan words and some more general synthesizing).The translations of some Tibetan texts are also missing.
The author has integrated the data of a new article (by Jue Liang & Andrew Taylor) and strongly criticizes it. I totally agree with her criticisms but not with the way of presenting them. In my view, the critics should not fit into a new section (Direction for Future Research, p. 17), but either in the introduction or in relevant chapters of her article. The whole chapter is in my view superfluous and parts of it could also go into the conclusion, which is a bit short.
Overall it is also necessary to take out some repetitions. I tried to mark a maximum, but there might be more.

Author Response
Thanks for your suggestions ad thoughtful review.
I rewrote some words accrediting your first review. Please see the PDF attached here.
Following your second review, I worked on these:
(1) I changed the italics for towns, regions, names, and the part of Tibetan laguage.
(2) I put the criticisms in part 2 and rewrote a few sentences.
(3) I deleted the title of Future research and combined the Conclusion and Future Research.
Thanks again! Please see the attachment.
Padma
